# Optimizing XGBoost Performance for Fish Weight Prediction through Parameter Pre-Selection

Mahdi Hamzaoui [1,†,‡] , Mohamed Ould-Elhassen Aoueileyine [1,‡] , Lamia Romdhani [2,*] and Ridha Bouallegue [1]

1 Innov'COM Laboratory, Higher School of Communication of Tunis, University of Carthage, Technopark Elghazala, Raoued, Ariana 2083, Tunisia; mahdi.hamzaoui@supcom.tn (M.H.); mohamed.ouldelhassen@supcom.tn (M.O.-E.A.); ridha.bouallegue@supcom.tn (R.B.)
2 Core Curriculum Program, Deanship of General Studies, Qatar University, Doha P.O. Box 2713, Qatar
* Correspondence: lamia.romdhani@qu.edu.qa
† Current address: Higher School of Communication of Tunis (SUPCOM), Technopark Elghazala, Raoued, Ariana 2083, Tunisia.
‡ These authors contributed equally to this work.

**Abstract:** Fish play a major role in the human nutritional system, and farmers need to know the accurate prediction of fish weight in order to optimize the production process and reduce costs. However, existing prediction methods are not efficient. The formulas for calculating fish weight are generally designed for a single species of fish or for species of a similar shape. In this paper, a new hybrid method called SFI-XGBoost is proposed. It combines the VIF (variance inflation factor), PCC (Pearson's correlation coefficient), and XGBoost methods, and it covers different fish species. By applying GridSearchCV validation, normalization, augmentation, and encoding techniques, the obtained results show that SFI-XGBoost is more efficient than simple XGBoost. The model generated by our approach is more generalized, achieving accurate results with a wide variety of species. Using the r2_score evaluation metric, SFI-XGBoost achieves an accuracy rate of 99.94%.

**Keywords:** aquaculture; fish weight; machine learning; artificial intelligence

**Key Contribution:** In this work, we propose a novel approach SFI-XGBoost that optimizes parameter pre-selection after fish weight prediction, resulting in highly accurate fish weight estimation for a diverse range of fish species. By introducing this innovative method, we significantly enhance the applicability and precision of fish weight prediction models, addressing a critical need in the field of fisheries research and management.





## 1. Introduction

Aquaculture has become a major part of the food production chain. It plays an important role in ensuring food security in many developing countries. The world production of farmed fish has reached a record 82.1 million tons, which is about 52% of the total production of fish for human consumption [1].

Fish is a key component of the human diet worldwide. It is an excellent source of nutritious protein. In intensive fish farming, fish weight estimation is very important for aquaculture industries. It assists the farmer in having a clear prediction level of fish growth and therefore forecasts future production time and cost. Accurate, reliable, and affordable prediction methods can optimize feeding management and ensure that the fish receive adequate nutrients for their growth [2]. This can help avoid under- or over-feeding, which can have negative effects on fish growth and health [3,4], and therefore maximize farm yields, improve profitability, and enhance the environmental sustainability of aquaculture. On the other hand, by continuously monitoring fish growth, farmers can quickly identify signs of disease or health problems and address them efficiently [5].

The majority of the current techniques to estimate fish growth are based on biometries directly measured on a subsample, which are very stressful to the fish. The involvement of manpower is necessary to perform these measurements manually [6]. Older methods of predicting fish weight often include the use of body measurements to estimate total weight [7]. Commonly used methods include measuring the total length of the fish, its height, or the girth, which is the widest part of the fish. Weight calculations are then made according to formulas appropriate to each species, which is based on the proportional relationship between height and length [8–10]. Applying the same formula to another species of fish may give inaccurate results because of their different shapes.

Newer methods of predicting fish weight can be more accurate and take into account a variety of factors to provide more precise and individualized predictions. Machine learning is a useful solution for predicting fish weights. The models generated can be applied to a wide range of fish species.

In this paper, we describe a study that uses machine learning techniques to predict fish weight. This is a new approach that generates a generalized model that can be applied to a wide variety of fish types. To provide a comprehensive overview, studies conducted earlier on the same subject and their results are discussed in the Section 2. In addition, the Section 3 provides theoretical definitions of the different methods used in this work. The Section 4 describes the data acquisition phase and outlines the data preprocessing methods used in the study. This section compares different algorithms for estimating fish weight and introduces the novel SFI-XGBoost approach. The ensuing section, Section 5, reports the study's findings and evaluates the performance of SFI-XGBoost compared to previous approaches. Finally, the paper concludes in the Section 6, which presents a forward-looking vision and insight into future works.

## 2. Related Work

Fish weight prediction is an important area of research in aquaculture. With the increasing demand for fish worldwide, the accurate prediction of fish weight has become crucial for fish stock management and production planning. In this context, many studies have been conducted to predict fish weight using machine learning techniques. In a recent study, Yunhan Yang et al. compared several methods as linear regression, power regression model, k-nearest neighbor, ridge regression, decision tree, random forest, gradient boosting, and multilayer perceptron for predicting fish weight. By applying these methods on four fish species, they showed that genetic programming for symbolic regression is the best method. It consists of integrating more features to the model in order to determine the hidden relationships [11]. Hui Li et al. proposed a new approach to predict fish weight in order to optimize the feeding system. They first implemented a model for the feeding system. A second fish weight prediction model was developed. It uses linear regression algorithms and integrates the feeding model. The parameters of the feeding model are adjusted according to the fish growth evolution [12].

Many other studies have used computer vision techniques to predict fish weight from an image. Raihan Islamadina et al. began by manually collecting the lengths, widths and heights of five examples of tuna. They manually measured the weights of the fish. The formula for calculating the weight of the fish from the length, width and height measurements was obtained. The calibration and segmentation processes for these fish images were carried out by the system. The feature extraction operation was carried out to determine the measurements made by the computer. The comparison between the actual weights of the fish and the result given by the system gave an error rate equal to 5.66% [13]. In this study, the authors worked on five samples only. This small number may influence the accuracy of the results obtained. Data augmentation techniques can improve the accuracy of this work. The calibration methods used may not be able to resolve the problems associated with the depth of the fish in the tank. To predict the weight of flatfish, K. H. Hwang et al. compared two different methods. The first one uses the standard equation given by the National Fisheries Research and Development Institute NFRDI

$Y = 0.2473L^{2.0247}$. The second method uses the vision image to determine the fish area. A linear regression model is then established to calculate the weight from this area [14]. This study was designed to predict the weight of a single species, the flatfish. The high linearity between the area and the weight of the fish enabled an accuracy rate of 99.72% to be achieved. Prediction using different machine vision techniques and CNN is very efficient. Dmitry A. Konovalov et al. applied CNN on two instances. The first one was trained on 200 fish masks that had been manually segmented but with the fins and tails removed. The second one received training with 100 whole-fish masks. The results of this study prove that the error rates did not exceed 4.36% [15]. The three deep CNN architectures VGG-11, ResNet-18 and DenseNet-121 were used by Yunhan Yan et al. to predict the weight of Australasian Snapper fish from images. After training the models, the results showed the efficiency of the three architectures with an accuracy rate that reached 0.96 [16]. This study opts for the approach of predicting fish weight directly from an image. Image quality is a key element in this method. Efforts should be made in the image-processing task to ensure a better result. Naruephorn Tengtrairat et al. estimated the weight of tilapia fish using deep learning and linear regression algorithms. The age of the fish was considered in this study as an important feature in the prediction process [6]. Measuring fish characteristics manually was considered by Nicolò Tonachella et al. as a costly and stressful operation for marine species. Using computer vision and linear regression models, a system for estimating the length and weight of fish was implemented in commercial aquaculture cage in the Mediterranean Sea. The results showed that the margin of error is ±1.15 [17].

The biomass of fish in aquaculture tanks is a very important element. It has a great effect on the growth of a fish. In order to make aquaculture companies more intelligent, researchers have been working on this factor. The objective of these studies is to control the space to ensure a maximum growth rate. Having the approximate number of fish can help the farmer to adjust the amount of feed and avoid food waste [18–20]. In another study, an intelligent management system for an aquaculture cage was implemented by Chung Cheng Chang et al., to visualize fish life and activities underwater. Using machine vision techniques, the display of fish length and weight is given in real time. This system can control the feeding system to reduce food waste. The movement of the fish can be used to detect which species are diseased [21]. The fish biomass estimation in ref. [22] used an Arduino board to measure the weights of live fish for on-land facilities or offshore cages. For determining fish weight at various growth stages, fish size is a crucially important element. Whereas special cameras are needed to capture free-swimming fish, machine vision offers an automated and efficient method for determining size.

Hongwei Qin et al. proposed an underwater system that can recognize the type of fish, and its different characteristics, such as height and length. This work combines neural networks with SVM algorithms for fish classification. This new method called deep fish reached an accuracy rate that exceeds 98% [23].

### 3. Background

#### 3.1. Min-Max Normalization

*Min-max* normalization is the technique of transforming information that has been measured in engineering units to a number between 0 and 1. This makes comparing numbers that were acquired using different scales or measuring units simple [24]. The normalized value is described as follows:

$$MM(X_{i,j}) = \frac{X_{i,j} - X_{min}}{X_{max} - X_{min}} \tag{1}$$

#### 3.2. One-Hot Encoding

One-hot encoding is a common technique for managing categorical data. Categorical variables must be transformed into a format in order for ML models to be more effective at spotting instances of insider data leaking [25].

### 3.3. Machine Learning Regression Algorithms

3.3.1. Linear Regression

In linear regression, the observed data are used to fit a linear equation that seeks to model the connection between two variables. A simple linear regression model is shown as follows:

$$y_i = x_{i1}w_1 + \ldots + x_{iD}w_{1D} + e_i \tag{2}$$

where $i = 1, 2, \ldots, M$ signifies the number of observations, $x_i \in RD$ stands for the explanatory variable, $y_i$ is the dependent variable, $w \in RD$ is the regression coefficient, $D$ means the regression model order with $M > D$, and $e_i$ is the observation error [26].

3.3.2. Ridge Regression

Ridge regression is regarded as a very helpful approach for solving the multicolinearity issue. One of least squares, subject to a particular kind of parameter restriction, serves as its formal formulation. An over-determined system of linear equations is typically solved by the linear least squares method, $Y = X\beta$, which aims to reduce the residual: $\|Y - X\beta\|^2$, where $Y$ is the $n \times 1$ vector, $X$ is the $n \times p$ matrix ($n \geq p$), $\beta$ is the $p \times 1$ vector, and $\|\|$ is the Euclidean norm [27].

3.3.3. Decision Tree

In order to build a model that predicts the value of the output variable based on the input variables in the feature vector, decision trees are efficient regression algorithms. The tree is notable for its quick execution speed, simplicity in rule interpretation, and scalability for huge multidimensional datasets. The main goal of building the tree is to divide the training set examples into subsets of almost all examples belonging to the same class. Only one attribute is used by each decision rule at a time [28].

3.3.4. Random Forest

One of the well-known learning algorithms, the random forest algorithm (RF), is considered an extremely superior and powerful technique in a machine learning system for classification problems. The key concept is to use RF classification on a recommendation system to identify the best related products based on user preferences [29]. The RF regression model is presented in Equation (3):

$$h(x) = \frac{1}{P} \sum_{n}^{p=1} h(y, \lambda p) \tag{3}$$

3.3.5. eXtreme Gradient Boosting (XGBoost)

One of the robust boosting techniques in the machine learning system is XGBoost. Based on the data structure, this algorithm has maximal accuracy when it comes to forecasting, classifying, and optimizing the stated system. This system's recommended products combine the categorization and prediction results by employing the collaborative filtering technique. The item-based collaborative filtering technique predicts the rate of clicked products while simultaneously assessing the rate of neighboring products. The target product is also at the same stage of recommendation to the user if the neighboring products are extremely similar to the user's tastes. The variations in product prediction weight are useful for improving prediction and recommendation outcomes [29]. To determine the likelihood of potential product projections, we must first obtain the actual values, which are defined as Equation (4):

$$[P_\alpha, P_{1-\alpha}] \tag{4}$$

*3.4. Metric Evaluation Models*

3.4.1. MAE (Mean Absolute Error)

MAE is the basic evaluation metric; it is used to compare the advantages and disadvantages of different algorithms [30]:

$$MAE = \frac{1}{N} \sum_{i=1}^{N} |Y_i - \overline{Y_i}| \tag{5}$$

3.4.2. MSE (Mean Square Error)

If there are any outliers that need to be found, MSE can be employed. Due to the $L_2$ norm, MSE is really excellent at assigning higher weights to such points. It is obvious that if the model ultimately produces a single extremely poor prediction, the squaring portion of the function magnifies the error [31]:

$$MSE = \frac{1}{m} \sum_{i=1}^{m} (X_i - Y_i)^2 \tag{6}$$

(best value = 0; worst value = $+\infty$).

3.4.3. R2_Score (Coefficient of Determination)

This is a typical regression model metric that quantifies how well the predicted values match the actual data points:

$$r^2 = 1 - \frac{RSS}{TSS} \tag{7}$$

where $r^2$ coefficient of determination; $RSS$ is the sum of squares of residuals; and $TSS$ is the total sum of squares. The performance of the linear regression model serves as the basis for creating the benchmark $r^2$ score, training time, and predicting time. The simplest regression model is linear regression. It is first determined whether the data exhibit a linear trend or not because Occam's Razor advises using the simplest model [32].

**4. Methodology**

In this section, we present the methodology employed for developing the novel hybrid approach for fish weight prediction. Our proposed approach leverages the strengths of multiple techniques to enhance prediction accuracy and capture complex relationships within the dataset. The methodology can be summarized in the following steps.

*4.1. Data Collection*

The Fish Market dataset is a comprehensive and meticulously curated collection of fisheries and aquaculture data, designed to facilitate research and analysis in the field of marine resource management, biodiversity conservation, and market trends. This dataset is accessible through Kaggle and provides a valuable resource for researchers, analysts, and policymakers seeking to understand various facets of the global fish market [33]. It has been sourced and compiled by experts in the field, ensuring its reliability and relevance for a wide range of studies.

Dataset Composition

The Fish Market dataset comprises a diverse set of attributes and features related to various fish species. These features include but are not limited to the following:

- Species: The specific fish species under consideration.
- Weight: The weight of the fish, typically measured in grams.
- Length1: The length of the body from the tip of the mouth to the base of the caudal fin, along its dorsal side, measured in centimeters.

- Length2: The measurement from the tip of the fish's mouth to the tip of the tail fin along a diagonal line, measured in centimeters.
- Length3: This is the length of the line from the upper point of the tail to the lower point of the mouth, measured in centimeters.
- Height: The height of the fish, also measured in centimeters.
- Width: The width of the fish, measured in centimeters.
- Market Region: The geographic region or market where the fish was recorded.
- Market Category: The category of the market, such as retail or wholesale.

The dataset contains 159 lines, each line representing the specific measurements of the fish. The fish in the dataset are European Bream, Roach, WhiteFish, Common Perch, Northern Pike and Delta Smelt. The data are distributed as shown in Table 1.

**Table 1.** The label distribution in the dataset.

| Fish Species | Scientific Name | Count of Samples | Fish Shapes |
|---|---|---|---|
| European Bream | Abramis brama | 35 | |
| Roach | Rutilus rutilus | 20 | |
| WhiteFish | Coregoninae | 17 | |
| Common Perch | Perca fluviatilis | 56 | |
| Northern Pike | Esox lucius | 17 | |
| Delta Smelt | Hypomesus olidus | 14 | |

*4.2. Data Preprocessing*

The preprocessing operations were applied on the data in order to make the model more efficient in the training phase. Using the data normalization technique, all the data were converted to values between 0 and 1. A one-hot encoder is used to encode labeled fields. An outlier analysis is performed to remove the outliers. As mentioned in Table 1, the distribution of data is not equitable across the different classes. The synthetic minority over-sampling technique (SMOTE) is used to create new samples while preserving the data structure. The aim of this technique is to increase the number of samples in classes with low numbers. Finally, a data augmentation technique was applied to the dataset to increase the number of records to 421.

*4.3. Model Selection and Training*

To establish a robust prediction model for fish weight in aquaculture contexts, it is essential to undertake a thorough algorithmic comparison. This study involves evaluating the performance of five regression algorithms linear regression, ridge, decision tree, random forest, and XGboost on a standardized and preprocessed dataset. The dataset, having undergone meticulous cleaning and preprocessing, ensures data quality and consistency across the analyses. To gauge the efficacy of these algorithms, three key evaluation metrics, mean square error (MSE), mean absolute error (MAE) and r2_score, are employed. These metrics collectively quantify predictive accuracy and model fit, offering a comprehensive assessment of the algorithmic performance. Through this systematic examination, the goal is to discern the algorithm that exhibits superior predictive prowess for fish weight in aquaculture. Ultimately, the findings provide valuable insights for selecting an optimal algorithmic approach in the realm of aquaculture prediction modeling.

### 4.4. Multicollinearity Analysis Methods

The significance of methods for analyzing multicollinearity within datasets cannot be overstated. These methods enable the identification of variables with substantial correlations among them. By isolating such variables, it becomes feasible to enhance model clarity. Eliminating one of these correlated variables has the potential to enhance model quality significantly.

#### 4.4.1. Variance Inflation Factor Method (VIF)

A statistical tool for evaluating the level of collinearity between independent variables in a linear regression model is the variance inflation factor (VIF). The VIF calculates the increase in variance in a regression coefficient estimate as a result of collinearity between an independent variable and other ones. The variance of the regression estimator of an independent variable in the full model is compared to the variance of the regression estimate of the same one in a basic model to obtain the VIF (including only that variable). According to tradition, a high VIF implies a significant correlation between the variables and could point to a collinearity issue in the model. To reduce collinearity and raise the model's goodness of fit, variables with high FIVs can be eliminated from the model.

#### 4.4.2. Pearson's Correlation Coefficient Method (PCC)

Correlation analysis is used to assess the degree of a link between two item sets. The strength can be calculated from the direction, form, and dispersion strength. The correlation coefficient is frequently employed to express this relationship in numerical terms. The correlation coefficient is estimated within a particular predetermined range, depending on the algorithm. Based on the coefficient's value within the given range, it is feasible to determine the coefficient's strength and direction. A positive sign for the coefficient indicates a positive correlation between the two variables, whilst a negative value for the coefficient indicates a negative correlation. Using the Pearson's correlation coefficient approach, a statistical analysis of the collinear relationship between two variables is performed. It includes the ratio of co-variance as well as the standard deviation of the data values between the two given variables. Think about the variables *A* and *B*. Pearson's correlation coefficient can then be calculated using the formula below:

$$C_{A,B} = \frac{covariance(A,B)}{\sigma_A \sigma_B} \tag{8}$$

where $C_{A,B}$ is the correlation coefficient, $covariance(A,B)$ is the covariance, and $\sigma_A$ and $\sigma_B$ are the standard deviations of *A* and *B*, respectively.

#### 4.4.3. Select Features Importance Method (SFI)

SFI is a feature selection method developed in this study. It uses the two multicollinearity analysis methods VIF and PPC to remove features of lesser importance in predicting fish weight. It selects only those features that have low values of multicollinearity with the feature weight according to both VIF and PCC. Features that have high collinearity with the feature weight according to both the VIF and PCC methods are automatically removed from the prediction process.

### 4.5. SFI-XGBoost Approach

As shown in the diagram in Figure 1, SFI-XGBoost is a hybrid approach that starts by applying the SFI method, a new approach that combines the two methods VIF and PCC. VIF consists in evaluating the multicollinearity between the explanatory variables of the XGBoost model. PCC allows to determine the importance of an attribute in the prediction of a target. Our SFI method crosschecks the two methods VIF and PCC in order to return the most important features selected by VIF and PCC both. Then, it returns an array containing only the features that are important for the prediction process. SFI-XGBoost trains with

the new set of input features, while applying the GridSearchCV method, which stands for grid search cross validation. GridSearchCV is a technique used in machine learning to tune the hyperparameters of a machine learning model in order to find the best combination of hyperparameters for a given task. It is a systematic way to search through a predefined set of hyperparameter values to determine which combination results in the best performance for a specific machine learning algorithm.

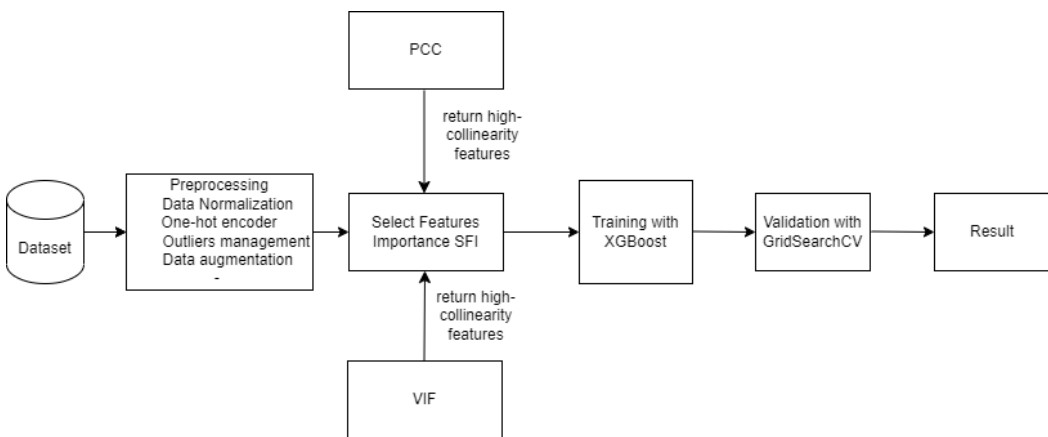

**Figure 1.** SFI-XGBoost flowchart.

## 5. Results and Discussion

### 5.1. Algorithms Evaluation

The evaluation of the algorithms using MSE, MAE and r2_score metrics showed that the linear regression performance is medium; the values of MSE/MAE and r2_score found are 0.0407/0.1649 and 94.52%, respectively. Ridge's results are also medium, the values of MSE/MAE and r2_score found being 0.0385/0.1675 and 94.80%, respectively. The decision tree and random forest algorithms had MSE/MAE and r2_score values between 0.0224 and 0.0163, between 0.0998 and 0.0795, and between 97.01% and 97.76%, respectively, which proves that these two algorithms are very efficient in predicting fish weight. The result with XGBoost is more accurate than the other models; the obtained values of MSE/MAE and r2_score are, respectively, 0.0129/0.0745 and 98.24%. The following Table 2 illustrates the evaluation of the different models compared.

**Table 2.** Model evaluations.

| Model | MSE | MAE | r2_Score (%) |
|---|---|---|---|
| Linear Regression | 0.0407 | 0.1649 | 94.52 |
| Ridge | 0.0385 | 0.1675 | 94.80 |
| Decision Tree | 0.0224 | 0.0998 | 97.01 |
| Random Forest | 0.0163 | 0.0795 | 97.76 |
| XGBoost | 0.0129 | 0.0745 | 98.24 |

The results of the fish weight predictions show the effectiveness of the XGBoost algorithm compared to the other algorithms. Figure 2 clearly shows that Xgboost achieves better prediction than linear regression (LR). In the rest of this study, XGBoost is chosen as a suitable method for our prediction process.

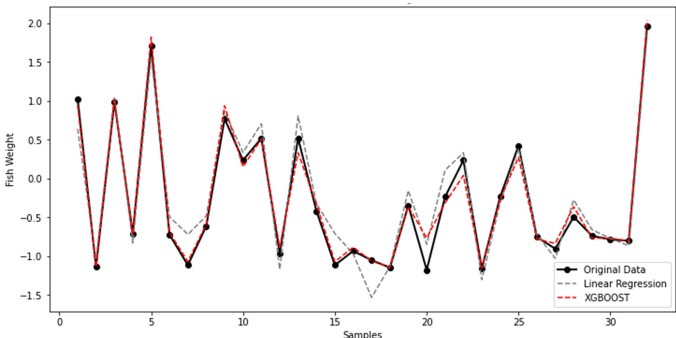

**Figure 2.** The accuracy visualization of the XGBoost and LR algorithms.

*5.2. Data Analysis*

Applying the VIF method, the results in Table 3 show that the height, width and Length2 characteristics have a low correlation with the target variable. On the other hand, the Length1 and Length3 variables are strongly correlated with the target weight. We can deduce that, according to VIF, the presence of height, width and Length2 can improve the quality of the model, whereas the presence of Length1 and Length3 can lead to accuracy and stability problems during training. Analysis of the PCC method shows in Figure 3 that, compared with the other characteristics, Length1, Length2 and Length3 have a strong correlation of 0.92 with weight, which is a target characteristic. According to the PCC method, eliminating the remaining height and width features can lead to more accurate model prediction. In this study, the SFI method implements the select_feature_importance() Python function, which allows to remove features recommended by both PCC and VIF. In our case, as shown in Figure 4, SFI removes Length1 and Length3 and keeps Length2, height and width in the list of characteristics. From this new SFI approach, we deduce that the parameters that are used to enable good prediction of the weight of the fish are its diagonal length (Length2), height and width. The same approach shows that the vertical length (Length1) and the crossed length (Length3) of the fish are not very useful in the weight prediction phase.

**Table 3.** The variance inflation factor VIF of the different features.

| Features | VIF |
|---|---|
| Length3 | 2559.12 |
| Length1 | 2373.50 |
| Length2 | 136.50 |
| Width | 84.46 |
| Height | 74.50 |

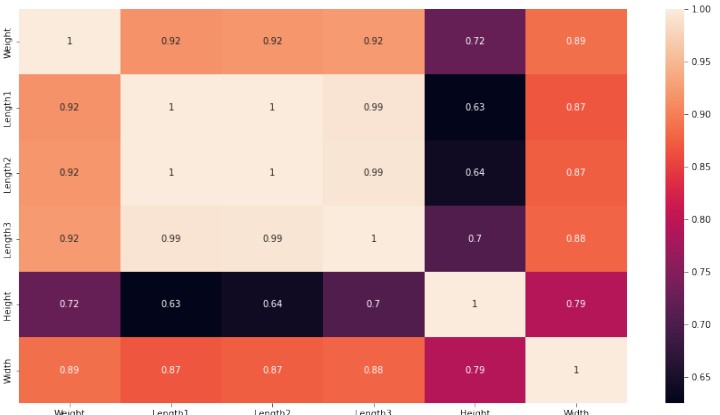

**Figure 3.** Pearson's correlation coefficient analysis.

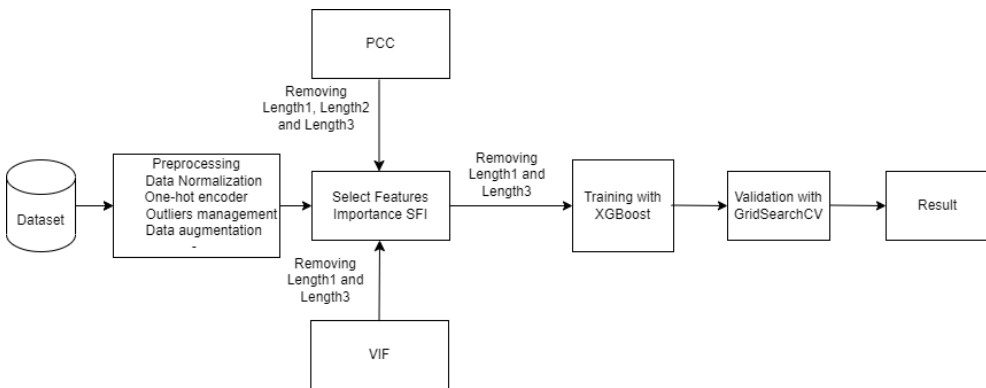

**Figure 4.** The SFI-XGBoost approach.

In addition, from the curves illustrated in Figure 5, the results clearly show that the multicollinearities Length1/Weight and Length3/Weight are very strong. This visualization defends the SFI approach much more, which eliminates the Length1 and Length3 features from the dataset. It is considered a disturbing attribute in the training phase of our model.

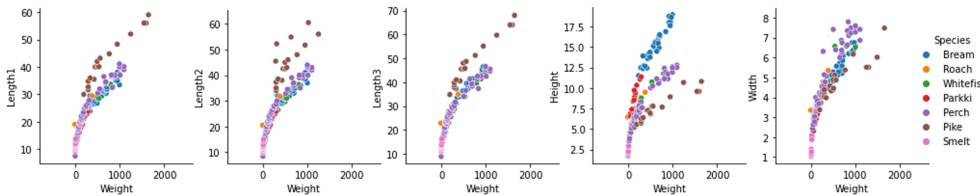

**Figure 5.** The feature multicollinearities.

### 5.3. SFI-XGBoost Performances

In the new approach SFI-XGBoost, XGBoost is combined with the two attribute selection methods VIF and PCC in order to improve its performance. Several machine learning techniques were also applied. To adjust the hyperparameters and the model efficiency, the cross-validation technique using GridSearchCV method was implemented. The results presented in Table 4 and Figure 6 show that this new approach is more efficient than a simple XGBoost model.

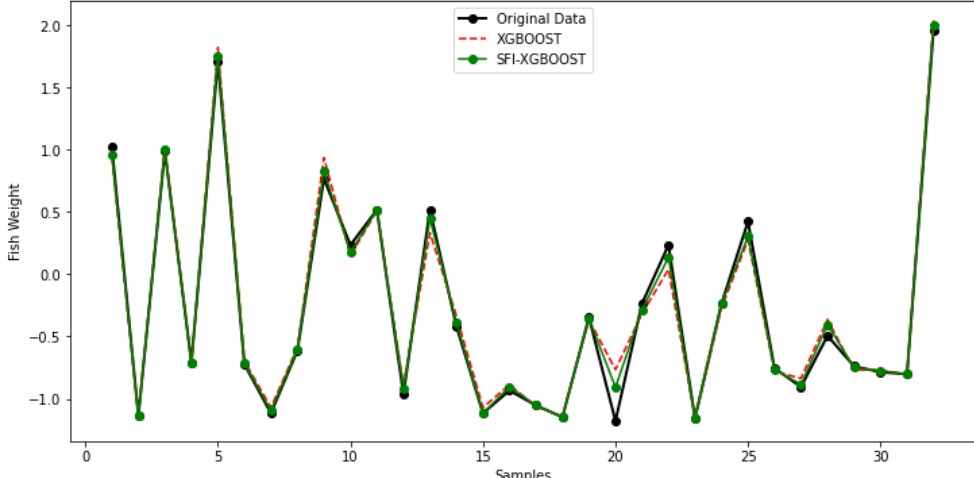

**Figure 6.** The XGBoost and SFI-XGBoost performance visualizations .

**Table 4.** Table of comparison between XGBoost and SFI-XGBoost performance results.

| Model | MSE | MAE | r2_Score (%) |
|---|---|---|---|
| XGBoost | 0.0129 | 0.0745 | 98.24 |
| SFI-XGBoost | 0.0006 | 0.0175 | 99.94 |

*5.4. Further Discussion*

To evaluate the performance of the new hybrid SFI-XGBoost approach, the results of this study are compared with those of several other similar studies in Table 5. An analysis is made to evaluate the methods used, the data characteristics, the performance indicators and the results obtained. Based on these comparisons, the advantages and limitations of these fish weight prediction methods are elaborated.

After comparing our study with other similar works, we can conclude that all these studies estimate the weight of one type of fish. That is, they are not applicable in a polycultural context. Our new approach is broader and can estimate weights of different fish species farmed in the same environment. In addition, the majority of existing studies usually rely on a single attribute to make the prediction. However, our work develops a new approach that allows us to adjust the set of features that are eventually useful in the learning phase. This process allows us to have a preferential model that reaches an accuracy rate of 99.94%. One of the main limitations of our study is related to the quality and quantity of the data used to train the models. Indeed, the dataset contains some data that are biased or incomplete. To overcome this issue, the use of other pre-processing techniques, such as data normalization, one-hot encoder, outlier analysis and data augmentation, allow us to improve the quality and volume of the data to have a powerful model.

**Table 5.** Table of comparisons between this work and related studies.

| Reference | Method | Data Characteristics | Fish Species | Performance Indicators |
|---|---|---|---|---|
| [14] | The flatfish is characterized by its large surface area. After capturing the flatfish images, preprocessing operations were applied. Using image segmentation techniques, the tail was removed to obtain a more accurate estimate of the area. Using the formula $W = 3.674\,A - 2.145$, where $W$ is the weight and $A$ is the area, the weight is estimated. | Flatfish Images captured. | Flatfish. | r2_score: 99.72% |
| [16] | The method used in this study is to train the convolutional neural network model with the external dataset. This training is performed with three different architectures of CNNs: VGG-11, ResNet18 and DenseNet-121. The results obtained with DenseNet-121 are the best. | A dataset containing images of fish and their weights which is produced by PFR. | Australasian Snapper. | r2_score: 96% |
| [34] | This method consists of collecting data (length and weight of fish) manually. According to the data found, the equation $W = 0.0196L^{2.9868}$ is established, where $W$ is the weight and $L$ is the length. Then, the length of the fish is determined from its image. Finally, the weight is estimated using the equation established at the beginning. | Fish lengths and fish weights are collected manually. Images of fish are captured to estimate the weight. | Red Tilapia. | ACC: 93.01% |
| [35] | An IoT system is used in this work. The collection of fish images is done from two positions (two cameras). The images are sent via a LoRa network to a web application. Fish image-processing operations are applied to determine the length. The weight estimation is made using the polynomial regression formula $W = 0.1017L^3 - 4.8944L^2 + 93.44L - 583.06$, where $W$ is the weight and $L$ is the length. | Fish images are captured from two cameras in an IoT system including a Raspberry Pi board, LoRaWAN and a web application. | Nile Tilapia. | Mean Percentage Error: 2.82% |

**Table 5.** *Cont.*

| Reference | Method | Data Characteristics | Fish Species | Performance Indicators |
|---|---|---|---|---|
| [36] | In this study, a system composed of several hardware and software components is implemented. The detection of fish images is performed with videos captured with a NIR camera. The Haar Classifier is a tool that detects a fish object in real time. After processing the image, the detection of length and width allows to estimate the weight of the fish using the equation $W = 1.861*10^{-8}*L^3$, where W is the weight and L is the length. | Videos captured with a NIR camera from an experiments tank. | Tilapia. | ACC: 92%. |
| This work | Our work consists of developing a new approach, SFI-XGBoost. It selects the most important features before passing them to XGBoost for training. SFI-XGBoost is based on the features Length1 Length2, Length3 and width to create the model. This model is applicable to several fish species. | The dataset is retained from the Kaggle web site. It contains the attributes species, Length1, Length2, Length3, width, height and weight. The number of records in this dataset is 159. | European Bream, Roach, WhiteFish, Common Perch, Northern Pike and Delta Smelt | r2_score: 99.94% |

Table 6 describes the application of some similar approaches on the dataset used in this work. Using the formulae proposed by the three studies [34–36] to predict fish weight gave scores of 51.74%, 63.17% and 50.52%, respectively. We can deduce that the scores are too low and that the models are not efficient when applied to a variety of fish types. They are all designed for a single type of fish or fish of similar shape. These models will not be useful in the context of predicting the weights of fish of different shapes. Our model is designed to predict the weight of a variety of fish and achieves a score of 99.94%.

**Table 6.** Table evaluating the effectiveness of the different approaches on the dataset from this work.

| Reference | Proposed Formula for Calculating Fish Weight | Fish Species | r2_Score |
|---|---|---|---|
| [34] | $W = 0.0196L^{2.9868}$ where W: Weight and L: Length | Red Tilapia | 51.74 % |
| [35] | $W = 0.1017L^3 - 4.8944L^2 + 93.44L - 583.06$ where W: Weight and L: Length. | Nile Tilapia | 63.17 % |
| [36] | $W = 1.861 * 10^{-8} * L^3$ where W: Weight and L: Length. | Tilapia | 50.52 % |
| This work | SFI-XGBoost approach | European Bream, Roach, White-Fish, Common Perch, Northern Pike and Delta Smelt | 99.94% |

## 6. Conclusions and Future Work

Fish weight prediction is very important for the development of sustainable aquaculture. It helps farmers to have an estimate of their production to know the optimal time of harvest, and to avoid overfeeding, improving, in the meantime, fish welfare. Based on an in-depth review of several issues in fish weight prediction, this study compared regression algorithms, such as linear regression, ridge, decision tree, random forest and XGBoost to conclude that XGBoost is the best among them. A new approach SFI for selecting features was implemented. It consists of selecting the most important features by cross referencing PCC and VIF. These were passed to XGBoost for training, validation and testing. Several machine learning techniques were applied on this new approach for optimization. An evaluation showed that the accuracy score evolved with SFI-XGBoost to reach a rate of 99.94%. All the tests implemented with SFI-XGBoost in this work show that this new approach is very effective. Based on our approach, we can deduce that the combination of length, width and height measurements gives a better prediction of fish weight without resorting

to area measurement. In the future, the selected algorithm will be tested on data collected directly in the rearing environment. Computer vision and image segmentation techniques will be integrated to predict the weight of fish during different time intervals. This will help to define the optimal time to harvest. This future system will be equipped with sensors to analyze water quality and other chemical and biological parameters.

**Author Contributions:** Conceptualization, M.H., M.O.-E.A., L.R. and R.B.; methodology, M.H., M.O.-E.A., L.R. and R.B.; software, M.H., M.O.-E.A., L.R. and R.B.; validation, M.H., M.O.-E.A., L.R. and R.B.; formal analysis, M.H., M.O.-E.A., L.R. and R.B.; investigation, M.H., M.O.-E.A., L.R. and R.B.; resources, M.H., M.O.-E.A., L.R. and R.B.; data curation, M.H., M.O.-E.A., L.R. and R.B.; writing—original draft preparation, M.H., M.O.-E.A., L.R. and R.B.; writing—review and editing, M.H., M.O.-E.A., L.R. and R.B.; visualization, M.H., M.O.-E.A., L.R. and R.B.; supervision, M.H., M.O.-E.A., L.R. and R.B.; project administration, L.R. All authors have read and agreed to the published version of the manuscript.

**Funding:** This research received the Open Access funding provided by the Qatar National Library.

**Data Availability Statement:** Not applicable.

**Acknowledgments:** Open Access funding provided by the Qatar National Library.

**Conflicts of Interest:** The authors declare no conflict of interest.

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
