# Peer review of "Optimizing XGBoost Performance for Fish Weight Prediction through Parameter Pre-Selection"

_fishes, doi:10.3390/fishes8100505_

Round 1

Reviewer 1 Report

The research has focused the fish weight prediction. As the result, SFI-XGBoost was more efficient and accurate than prior method, which is significant and valuable in aquaculture management, market analysis and economic study. The authors focused the machine learning, but there is still a lack of knowledge of aquatic biology and aquaculture. In general, it is an interesting research. But there are still some description not accurate and appropriate.

1 Line 70-71  "the" should be "The". The hyphen in " ac-cording" should be deleted.

2 Methodology I have read your dataset in Kaggle. If it is convenient, the link to this dataset should be added. And the specific information of the dataset should be clarify in the manuscript. Which species? The scientific name of fishes? How many fish? In which area/territory? These information are important to readers. Besides, there are several mistake in your dataset. For example, there are three length, but they are similar and confusing. The name bream, perch and so on are too vague. They are not accurate enough. Besides, authors have added a long section to describe the use case and benefits of the dataset. I don't think the description is necessary. How to measure the length, weight, height ACCURATELY not the application and the significance is the focus of this manuscript.

English Language is OK

Reviewer 2 Report

The manuscript titled "A New Hybrid Approach for Fish Weight Prediction" introduces a method for parameter selection, employing VIF and PCC techniques, to optimize the application of XGBoost, a machine learning algorithm, for fish weight prediction. The presented results are encouraging and demonstrate practical applicability. Notably, the authors have meticulously detailed their methodology in impeccable English, greatly enhancing the overall readability.

However, a major issue arises from the manuscript's lack of clear takeaways, which hampers the reader's understanding of the core contributions. For instance, the current title, "new hybrid approach," could benefit from greater specificity. A more informative title might be "Parameter Pre-selection Enhances the Performance of XGBoost in Predicting Fish Weight." I would like to emphasize that, as a non-native English speaker, my intention is to provide constructive feedback to enhance the manuscript's clarity and impact.

In all, it is crucial to underscore the key insights throughout the paper, ensuring that the reader can readily grasp the significance of the research.

Specific Points:

Lines 35-46: I understand that the previous methods have shown poor performance when dealing with fish of different body shapes and have not considered individual variations in fish growth and feeding. However, several expressions in this section appear to be similar. For example, in line 38, the authors mention "fish growth and feeding," while in line 43, they refer to "fish nutrient and growth." It would be helpful to clarify the differences between these terms and perhaps provide references for further context. Additionally, I became curious about how their method performed concerning these two aspects. Unfortunately, I found no answers in the manuscript. Please provide a clear response to this question.

Line 133: Please replace "com-paring" with "comparing."

Section 4.1.1: I recommend including information about the shape and species of fish within this dataset. Since in the later discussion (Section 5.4), the authors compare fish species, the results may shed light on how shape affects model performance.

Section 4.1.2: In my opinion, this section is not necessary.

Section 4.1.3: Please provide more detailed information. For instance, when you mention "a wide range," it would be helpful to specify the range statistically, summarizing the dataset.

Line 282: Please explain "GridSearchCV" and ensure consistent capitalization.

Line 315: Instead of "vertical length (Length3)," you might mean "(Length1)," I presume.

Line 344: The authors mention using "pre-processing techniques." However, parameter pre-selection could also be considered one of the pre-processing techniques. I suggest either removing the phrase "pre-processing techniques" or adding "other" before it to clarify that parameter pre-selection is one of several pre-processing techniques.

Lines 351-352: These are interesting points. While I comprehend that the authors' predictions outperform previous regression methods, the reasons behind this should be explained. As per the Introduction, it seems that fish shape plays a crucial role. Therefore, avoid terms like "different" or "unique." Instead, provide readers with insight into the types of shapes suitable for previous regression methods and those that still pose challenges, considering that the r2_score is not 100% perfect.

The quality of English language is good.

Reviewer 3 Report

The abstract should be more attractive. The implications of this study should be stated more clearly, because in this way only technical aspects of this research are detailed, As this journal is not just methodological, I guess the abstract should be implemented accordingly.

The paragraph from 24 to 28 should be removed. This paper is not a matter of the physical and chemical parameters affecting farmed fish growth.

The paragraph “Related work” should be rewritten, concepts are poorly explained, and the terminology is often very confusing.

I suggest to stress in the final conclusions which should be the best predictors of fish weight (lenght, area, a combination of measures?), as described in the results.

All other comments in the attached file.

English language is ok, some moderate changes are reported in the attached annotated pdf.
